# Redistribution of Weights and Activations for AdderNet Quantization

**Ying Nie**[1]  **Kai Han**[1,2]  **Haikang Diao**[3]  **Chuanjian Liu**[1]  **Enhua Wu**[2,4]  **Yunhe Wang**[1]*

[1]Huawei Noah's Ark Lab
[2]State Key Lab of Computer Science, ISCAS & UCAS
[3]School of Integrated Circuits, Peking University  [4]University of Macau
{ying.nie, kai.han, yunhe.wang}@huawei.com

## Abstract

Adder Neural Network (AdderNet) provides a new way for developing energy-efficient neural networks by replacing the expensive multiplications in convolution with cheaper additions (*i.e.*, $\ell_1$-norm). To achieve higher hardware efficiency, it is necessary to further study the low-bit quantization of AdderNet. Due to the limitation that the commutative law in multiplication does not hold in $\ell_1$-norm, the well-established quantization methods on convolutional networks cannot be applied on AdderNets. Thus, the existing AdderNet quantization techniques propose to use only one shared scale to quantize both the weights and activations simultaneously. Admittedly, such an approach can keep the commutative law in the $\ell_1$-norm quantization process, while the accuracy drop after low-bit quantization cannot be ignored. To this end, we first thoroughly analyze the difference on distributions of weights and activations in AdderNet and then propose a new quantization algorithm by redistributing the weights and the activations. Specifically, the pre-trained full-precision weights in different kernels are clustered into different groups, then the intra-group sharing and inter-group independent scales can be adopted. To further compensate the accuracy drop caused by the distribution difference, we then develop a lossless range clamp scheme for weights and a simple yet effective outliers clamp strategy for activations. Thus, the functionality of full-precision weights and the representation ability of full-precision activations can be fully preserved. The effectiveness of the proposed quantization method for AdderNet is well verified on several benchmarks, *e.g.* , our 4-bit post-training quantized adder ResNet-18 achieves an 66.5% top-1 accuracy on the ImageNet with comparable energy efficiency, which is about **8.5%** higher than that of the previous AdderNet quantization methods. Code will be available at `https://gitee.com/mindspore/models/tree/master/research/cv/AdderQuant`.

## 1   Introduction

Convolutional neural networks (CNNs) have achieved extraordinary performance on many vision tasks. However, the huge energy consumption of massive multiplications makes it difficult to deploy CNNs on portable devices like mobile phones and embedded devices. As a result, substantial research efforts have been devoted to reducing the energy consumption of CNNs [12, 16, 8, 7, 32, 37, 29].

Beyond pioneering model compression approaches, Chen *et al*. [4] advocate the use of $\ell_1$-norm instead of cross-correlation for similarity measure between weights and activations in conventional convolution operation, thus produces a series of adder neural networks (AdderNets) without massive

---

*Corresponding author

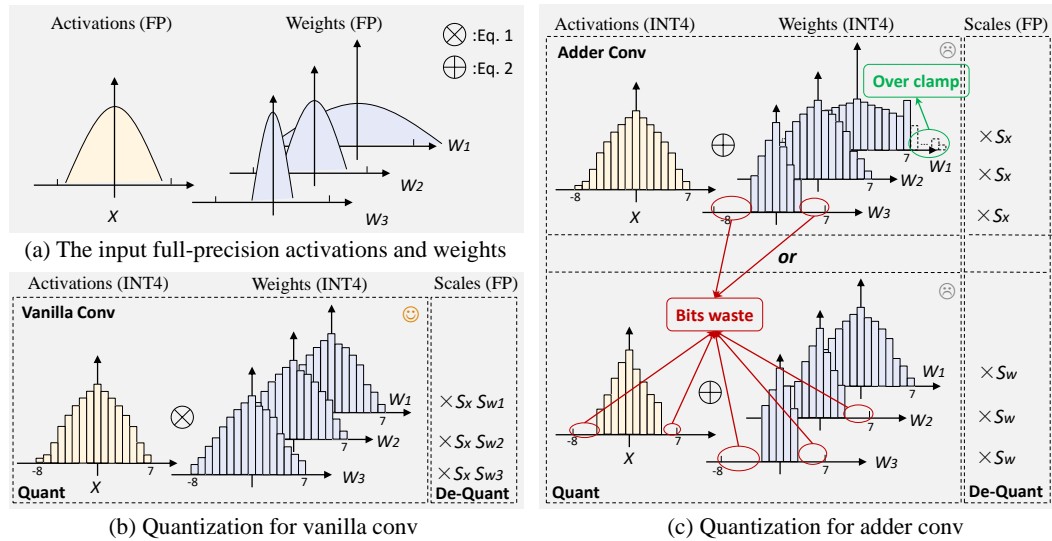

(a) The input full-precision activations and weights

(b) Quantization for vanilla conv

(c) Quantization for adder conv

Figure 1: Comparisons of quantization methods of vanilla convolution and adder convolution (symmetric 4-bit as an example). The independent scales adopted by vanilla convolution quantization can properly handle the challenge of large differences in weights and activations. One shared scale adopted in existing adder convolution quantization methods cannot handle this challenge.

number of multiplications. Compared with multiplication, addition is more energy efficiency, *e.g.* full-precision (FP) multiplication takes about $4\times$ energy as full-precision addition [15, 39, 33, 23]. AdderNets have shown impressive performance in large scale image classification [4, 38], object detection [5], *etc.* which shed light into the design of low-power hardware for AI applications.

The operation of low-bit addition has much lower energy consumption than the full-precision addition [15, 39, 33, 23]. To further improve the efficiency of AdderNet, low-bit quantization is a crucial option. Quantization algorithms have been studied in conventional convolution networks for a long time and achieved remarkable performance [20, 1, 19, 11, 28, 24, 18, 10, 36, 30]. As the law of commutation holds in the quantization process of multiplication, the scales of weights and activations for quantization can be independent of each other (Eq. 6). In practice, weights tend to be quantized with multiple scales based on the output channels, while activations are quantized with one scale based on the whole layer (Figure 1 (b)). However, the law of commutation does not hold in the $\ell_1$-norm quantization, which restricts the weights and activations in AdderNet to be quantized with shared scale (Eq. 7, 10). Therefore, existing AdderNet quantization techniques [35, 5] adopt only one shared scale to quantize both the weights and activations simultaneously. In practice, the shared scale is calculated based on the activations of the whole layer (Top of Figure 1 (c)) or the weights of the whole layer (Bottom of Figure 1 (c)). However, given a pre-trained full-precision AdderNet model, the ranges of weights usually vary widely between output channels, and the ranges of activations and weights also vary widely, both of which pose a huge challenge for quantization. If the range of activations is adopted to quantize weights, a large proportion of weights will be clamped, *i.e.*, over clamp. If the range of weights is adopted to quantize activations, a large proportion of precious bits will be never used, *i.e.*, bits waste. In both cases, a poor quantized accuracy will be incurred. In contrast, multiple independent scales adopted in the vanilla convolution can avoid this challenge.

To enhance the accuracy of quantized low-bit AdderNets, we first thoroughly analyze the problems of the one shared scale paradigm for quantization in existing AdderNet quantization methods. Then, we develop a novel quantization algorithm for AdderNet. Specifically, we first propose a scheme of group-shared scales with a negligible increase in computational workload to address the challenge that the ranges of weights vary widely between output channels. The pre-trained full-precision weights are divided into multiple groups by a quantization-oriented clustering process. Since the weights within a group are similar, the intra-group sharing and inter-group independent scales are thus used in our method. To further improve the performance of quantized adder neural network, we introduce a lossless range clamp scheme for weights to further reduce the difference on distribution of activations and weights in AdderNet. In practice, for groups where the range of the weights exceeds the range of the activations, we clamp the weights to the range of activations and then incorporate

the clamped values into the following *bias* term. Next, for those activations that contain outliers, a simple yet effective outliers clamp strategy for activations is presented to eliminate the negative impact of outliers. The extensive experiments conducted on several benchmarks demonstrate the effectiveness of the proposed quantization method.

## 2 Preliminaries and Motivation

In this section, we briefly revisit the necessary fundamental of adder neural network and existing quantization process for conventional CNN and AdderNet, then we describe the motivation.

### 2.1 Preliminaries

**Adder Neural Network (AdderNet).** Conventional convolution uses massive multiplications for computation, while adder convolution utilize addition to replace multiplication for reducing computational cost. Given the input activations $X \in \mathbb{R}^{H \times W \times c_{in}}$ and the weights $W \in \mathbb{R}^{d \times d \times c_{in} \times c_{out}}$, the traditional convolutional operation is defined as:

$$Y(m,n,c) = X \otimes W = \sum_{i=1}^{d} \sum_{j=1}^{d} \sum_{k=1}^{c_{in}} X(m+i,n+j,k) \times W(i,j,k,c), \tag{1}$$

where $H$ and $W$ are the height and width of activations, $c$ is the index of output channels, $d \times d$ denotes the kernel size, $c_{in}$ and $c_{out}$ are the number of input and output channels, respectively.

To avoid massive floating number multiplications, Chen *et al.* [4] advocate the use of $\ell_1$-norm instead of cross-correlation for similarity measure between activations and weights in CNN:

$$Y(m,n,c) = X \oplus W = -\sum_{i=1}^{d} \sum_{j=1}^{d} \sum_{k=1}^{c_{in}} |X(m+i,n+j,k) - W(i,j,k,c)|, \tag{2}$$

where $|\cdot|$ is the absolute value function. Since Eq. 2 only contains addition operation, the energy costs of the adder neural network can be effectively reduced [15, 33, 39]. With modified back-propagation approach and adaptive learning rate strategy, AdderNet achieves satisfactory performance on image classification [4, 38, 35], object detection [5], *etc*.

**Quantization for CNN.** The common uniform quantization for CNN linearly maps full-precision real numbers into low-bit integer representations, which is preferred by hardware efficiency. We take the symmetric mode without zero point as an example to describe the procedure of quantization. Given a full-precision input value $v$ and bit-width $b$, the quantized value $\bar{v}$ can be defined as:

$$\bar{v} = clamp(\lfloor v/s \rceil, q_n, q_p), \tag{3}$$

where the scale $s = 2 * max(|v|)/(2^b - 1)$, $q_n = -2^{b-1}$ and $q_p = 2^{b-1} - 1$. $\lfloor \cdot \rceil$ denotes the round function and $clamp(z, r_1, r_2)$ indicates that value $z$ is clamped between $r_1$ and $r_2$. Correspondingly, the de-quantized data can be computed by:

$$\hat{v} = \bar{v} \times s. \tag{4}$$

And the quantization loss can be defined as:

$$L_{quant} = |\hat{v} - v|. \tag{5}$$

Since the commutative law holds in multiplication, the multiplication operation of weights and activations in the conventional convolution can be converted as follows:

$$X \otimes W \approx \hat{X} \otimes \hat{W} = (s_x \bar{X}) \otimes (s_w \bar{W}) = (s_x s_w) \times (\bar{X} \otimes \bar{W}), \tag{6}$$

where $\otimes$ denotes the vanilla convolution operation in Eq. 1. In common settings [24, 19, 20, 1, 6], the weights adopt independent $s_w$ for each output channel, and all activations in a layer share one $s_x$. From Eq. 6, the data format of vanilla convolution operation is changed from full-precision to low-bit integer, which can reduce the energy consumption effectively.

**Quantization for AdderNet.** Due to the fact that the commutative law does not hold in adder convolution, the independent scales $s_w$ and $s_x$ cannot be adopted when quantifying weights and activations in adder convolution as we do in conventional convolution, that is:

$$X \oplus W \approx \hat{X} \oplus \hat{W} = (s_x \bar{X}) \oplus (s_w \bar{W}) \neq (s_w s_x) \times (\bar{X} \oplus \bar{W}), \tag{7}$$

where $\oplus$ denotes the adder convolution operation in Eq. 2. Therefore, the existing works on AdderNet quantization [35, 5] all adopt one shared scale $s$ for quantifying weights and activations. Thus, the quantized and de-quantized weights are defined as:

$$\bar{W} = clamp(\lfloor W/s \rceil, q_{wn}, q_{wp}), \quad \hat{W} = s\bar{W}, \tag{8}$$

similarly, the quantized and de-quantized activations are defined as:

$$\bar{X} = clamp(\lfloor X/s \rceil, q_{xn}, q_{xp}), \quad \hat{X} = s\bar{X}, \tag{9}$$

where $s = 2 * max(|X|)/(2^b - 1)$ or $s = 2 * max(|W|)/(2^b - 1)$. Therefore, the quantized adder convolution can be implemented:

$$X \oplus W = \hat{X} \oplus \hat{W} = (s\bar{X}) \oplus (s\bar{W}) = s \times (\bar{X} \oplus \bar{W}). \tag{10}$$

## 2.2 Motivation

Empirically, the values of weights and activations tend to vary widely in pre-trained full-precision AdderNet. We take the *layer1.1.conv2* in pre-trained adder ResNet-20 [14] as an example to visualize the statistics of the absolute ranges of activations and weights in Figure 2. Obviously, the absolute ranges of weights vary widely between output channels, and the absolute ranges of activations and weights also vary widely, both of which pose a huge challenge for the quantization with one shared scale. After thorough analysis, we conclude that when weights and activations in AdderNet are quantized with one shared scale, no matter whether the scale is calculated based on weights or activations, a large quantization loss (Eq. 5) will be incurred, resulting in a poor accuracy, especially in the case

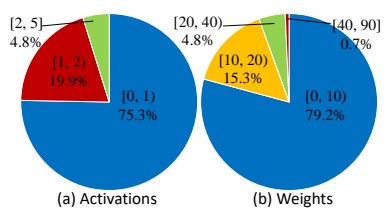

Figure 2: The statistics of the absolute ranges of activations and weights in the pre-trained full-precision AdderNet.

of low bits. For example, the Top-1 accuracy loss of 5-bit quantized adder ResNet-18 on the ImageNet is 3.3%, while the accuracy loss of 4-bit quantized adder ResNet-18 increases to 9.9% [35].

**Proposition 1.** *Denote $r_x$ and $r_w$ as the ranges of activations and weights in AdderNet, respectively. If $r_x$ is adopted to quantize weights, a large proportion of weights will be clamped. If $r_w$ is adopted to quantize activations, a large proportion of precious bits will be never used. In both cases, large quantization loss is incurred, resulting in a poor accuracy.*

*Proof.* For the first case where $r_x$ is adopted to quantize weights, according to Eq. 8, the detailed quantization of weights can be formulated as:

$$\bar{W} = clamp(\lfloor W(2^b - 1)/2r_x \rceil, -2^{b-1}, 2^{b-1} - 1). \tag{11}$$

Without losing generality, suppose $W = 50r_x$, then $\lfloor W(2^b - 1)/2r_x \rceil = 25(2^b - 1)$, if $25(2^b - 1)$ is clamped between $-2^{b-1}$ and $2^{b-1}$, then the proportion of the clamped value is $(25 * (2^b - 1) - 2^{b-1})/(25 * (2^b - 1)) = (49 * 2^{b-1} - 25)/(50 * 2^{b-1} - 25) \approx 0.98$. Besides, the quantization loss is $|\hat{W} - W| \approx |r_x - 50r_x| = 49r_x$, which will result in a significant accuracy degradation.

For the second case where $r_w$ is adopted to quantize activations, according to Eq. 9, the detailed quantization of activations can be formulated as:

$$\bar{X} = clamp(\lfloor X(2^b - 1)/2r_w \rceil, -2^{b-1}, 2^{b-1} - 1). \tag{12}$$

Similarly, suppose $X = 0.02r_w$, then $\lfloor X(2^b - 1)/2r_w \rceil = \lfloor 0.01(2^b - 1) \rceil$, if we clamp $\lfloor 0.01(2^b - 1) \rceil$ between $-2^{b-1}$ and $2^{b-1}$, the remaining $2^{b-1} - \lfloor 0.01(2^b - 1) \rceil \approx 0.98 * 2^{b-1}$ bits will be never used, which is a huge waste for precious bits. $\square$

## 3 Approach

In this section, we will describe the proposed quantization algorithm for AdderNet in detail. Firstly, we introduce the method of group-shared scales based on clustering. Then, the clamp for weights and activations are proposed, respectively.

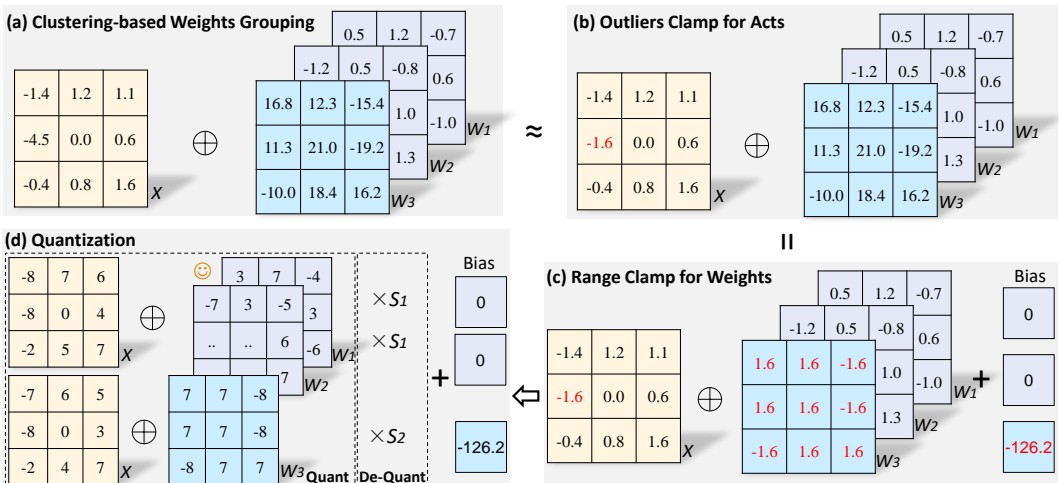

Figure 3: An illustration of the proposed quantization method for AdderNet (symmetric 4-bit as an example). The pre-trained full-precision weights are clustered into different groups. Then the clamp scheme for weights and activations are explored respectively to make efficient use of the precious bits and eliminate the negative impact of outliers.

## 3.1 Clustering-based Weights Grouping

To address the challenge that the ranges of weights in pre-trained full-precision AdderNet vary widely between output channels, we propose a novel quantization method with group-shared scales as illustrated in Figure 3 (a). Formally,

$$
\begin{aligned}
X \oplus W &= concat(o_1, ..., o_g), \\
\text{where} \quad o_j &= s_j \times (\bar{X} \oplus \bar{W}[:, :, :, \mathbb{I}_j]),
\end{aligned}
\tag{13}
$$

where $j = 1, ..., g$ is the index of groups, $g$ is the number of pre-defined groups. $\mathbb{I}_j$ indicates a set of indexes of weights belonging to the $j$-th group, and $s_j$ indicates the $j$-th scale. The set $\mathbb{I}$ is achieved by clustering the weights in the pre-trained full-precision model. Since the scale $s = 2 * max(|v|)/(2^b - 1)$, which indicates that $s$ and the maximum of $|v|$ are positively correlated. Thus, we cluster $v$ based on the maximum of $|v|$, not all the elements of $v$. In practice, the absolute maximum in each output channels of weights are taken as the clustering feature. Formally, the generated feature vector of weights is denoted as $[f_1, ..., f_{c_{out}}]$:

$$
f_c = \max(|W[:, :, :, c]|),
\tag{14}
$$

where $c = 1, ..., c_{out}$ is the index of output channels. Next, the feature vector is divided into pre-defined $g$ clusters $\mathbb{I} = \{\mathbb{I}_1, \mathbb{I}_2, ..., \mathbb{I}_g\}$. Any pair of points within each cluster should be as close to each other as possible:

$$
\min_{\mathbb{I}} \sum_{j=1}^{g} \sum_{c \in \mathbb{I}_j} |f_c - \mu_j|^2,
\tag{15}
$$

where $\mu_j$ is the mean of points in cluster $\mathbb{I}_j$. The method like $k$-means [13] can be adopted for clustering weights with the objective function in Eq. 15. Finally, the intra-group sharing and inter-group independent scales can be adopted for quantization. In practice, for groups where the range of the weights is within the range of the activations, the difference between the range of the weights and the activations is usually small, so the scale calculated based on the range of weights can be adopted for quantifying both the weights and activations. Conversely, for groups where the range of weights exceeds the range of activations, the range of weights and activations tends to differ significantly, making it difficult to quantize both weights and activations with one shared scale, which will be addressed in the next subsection.

**Analysis on Complexities.** Compared to quantifying weights and activations with only one scale, the proposed quantization scheme with group-shared scales can improve the performance of quantized AdderNet, but it will result in a small increase in Floating Point Operations (FLOPs). After thorough analysis, we conclude that the increased FLOPs is negligible.

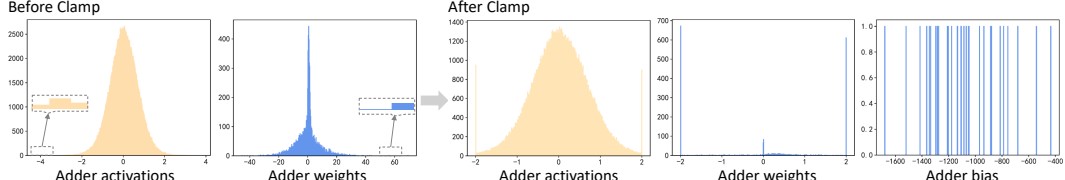

Figure 4: The distributions of activations and weights in pre-trained full-precision AdderNet.

**Proposition 2.** *Without losing generality, suppose $c_{out} = c_{in} = c$, then the additional FLOPs ratio introduced by the group-shared scales is approximately equal to $1/(6c + 1)$, which is negligible.*

The detailed proof is provided in the supplementary material under the assumption that $g = 4$, considering that the magnitude of $c$ is generally in the tens or hundreds for common networks, thus the additional FLOPs ratio is very small.

### 3.2 Clamp for Weights and Activations

To address another challenge that the ranges of activations and weights in pre-trained full-precision AdderNet vary widely, more specifically, the range of weights far exceeds the range of activations (Left of Figure 4), we present a lossless range clamp scheme for weights (Figure 3 (c)). Besides, a simple yet effective outliers clamp strategy for activations is also introduced to eliminate the negative impact of outliers (Figure 3 (b)).

**Range Clamp for Weights.** A lossless range clamp scheme for weights is first proposed. Formally, denote $W_c$ as any one output channel weights, *i.e.*, $W_c = W[:, :, :, c], c \in [1, ..., c_{out}]$, $r_x$ and $r_w$ as the ranges of activations and weights, respectively. In practice, we clamp the full-precision values of $W_c$ to the range of $[-r_x, r_x]$, then add the clamped values to the following *bias* $b$.

**Theorem 1.** *The proposed range clamp scheme for weights has no any effect on the accuracy, i.e., lossless. Formally,*

$$X \oplus W_c = X \oplus clamp(W_c, -r_x, r_x) + b,$$

$$where \ b = -\sum_{i=1}^{d}\sum_{j=1}^{d}\sum_{k=1}^{c_{in}} \max(|W_c[i, j, k]| - r_x, 0). \tag{16}$$

*Proof.*

$$X \oplus W_c = -\sum_{i=1}^{d}\sum_{j=1}^{d}\sum_{k=1}^{c_{in}} |X(m+i, n+j, k) - W_c(i, j, k)| \triangleq -\sum_{i=1}^{d}\sum_{j=1}^{d}\sum_{k=1}^{c_{in}} f(X, W_c(i, j, k)), \quad (17)$$

where $f(X, W_c(i, j, k))$ can be divided into three cases:

- if $W_c(i, j, k) > r_x$, then $f(X, W_c(i, j, k)) = (r_x - X(m+i, n+j, k)) + (W_c(i, j, k) - r_x)$.

- if $-r_x \leq W_c(i, j, k) \leq r_x$, then $f(X, W_c(i, j, k)) = |X(m+i, n+j, k) - W_c(i, j, k)|$.

- if $W_c(i, j, k) < -r_x$, then $f(X, W_c(i, j, k)) = (X(m+i, n+j, k) + r_x) + (-W_c(i, j, k) - r_x)$.

The above three cases can be unified into:

$$X \oplus W_c = X \oplus clamp(W_c, -r_x, r_x) - \sum_{i=1}^{d}\sum_{j=1}^{d}\sum_{k=1}^{c_{in}} \max(|W_c[i, j, k]| - r_x, 0). \tag{18}$$

$\square$

After the lossless range clamp process for weights, $r_w$ is equal to $r_x$. Thus, we can directly adopt $r_x$ to quantize the weights and activations.

**Outliers Clamp for Activations.** As shown in the left of Figure 4, some activations clearly have outliers. To avoid the negative impact of these outliers during quantization, we propose a simple yet effective outliers clamp strategy for activations. Specifically, the absolute activations are denoted as

$\mathbb{X} = \{|x_0|, ..., |x_{n-1}|\}$, and the sorted $\mathbb{X}$ in ascending order is denoted as $\widetilde{\mathbb{X}}$, where $n$ is the number of the values. We select the value $r_x = \widetilde{\mathbb{X}}[\lfloor \alpha * (n-1) \rceil]$ as the range of activations for the calculation of scale, where $\alpha \in (0, 1]$ is a hyper-parameter controlling the ratio of discarded activations. Similarly, there is another option, *i.e.*, $r_x = \widetilde{\mathbb{X}}[n-1] * \alpha$. However, this method can only reduce the negative impact of outliers, whereas ours can eliminate the negative impact of outliers.

## 4 Experiments

In this section, we conduct extensive experiments. Thorough ablation studies are also provided.

### 4.1 Experiments on CIFAR

**Re-train FP Networks.** Following the pioneering work of AdderNet [4], we first conduct experiments on CIFAR-10 and CIFAR-100 [21]. The CIFAR-10/100 dataset consists of 60,000 color images in 10/100 classes with $32 \times 32$ size, including 50,000 training and 10,000 validation images. The training images are padded by 4 pixels and then randomly cropped and flipped. We re-train the three relevant full-precision networks, including VGG-Small [3], ResNet-20 [14] and ResNet-32 [14], according to the experimental settings in AdderNet [4]. Specifically, the first and last layers of the network are kept as conventional convolution, and the other layers are replaced with adder convolution. We employ learning rate starting at 0.1 and decay the learning rate with cosine learning rate schedule. SGD with momentum of 0.9 is adopted as our optimization algorithm. The weight decay is set to $5 \times 10^{-4}$. We train the models for 400 epochs with a batch size of 256. The final results are provided in in the supplementary material, and are basically consistent with the results in [4]. The baseline results of CNN and binary neural network (BNN) [41] are cited from [4].

**Low-bit Quantization.** Following the general setting in quantization [31, 40, 22, 27, 25], we do not quantize the first and the last layers. The proposed quantization method can be employed in either post-training quantization (PTQ) or quantization-aware training (QAT). In practice, when the bits is 4, we begin to perform QAT procedure. Unless otherwise specified, the number of groups we use is 4. The hyper-parameter $\alpha$ controlling the ratio of discarded outliers in activations is empirically set to 0.999 in CIFAR experiments. In the QAT procedure, we keep the same settings as the FP training, except that the learning rate is changed to $1 \times 10^{-4}$, and only 50 epochs are used for saving the training time. The straight-through estimator [2] is adopted for avoiding the zero-gradient problem.

Table 1: Quantization results of AdderNets on CIFAR-10 and CIFAR-100 datasets.

| Model | Bits | CIFAR-10 (%) | | CIFAR-100 (%) | |
|---|---|---|---|---|---|
| | | PTQ | QAT | PTQ | QAT |
| VGG-Small | 8 | 93.42 (-0.02) | - | 73.56 (-0.04) | - |
| | 6 | 93.48 (+0.04) | - | 73.62 (+0.02) | - |
| | 5 | 93.41 (-0.03) | - | 73.50 (-0.10) | - |
| | 4 | 93.20 (-0.24) | 93.46 (+0.02) | 73.15 (-0.45) | 73.58 (-0.02) |
| ResNet-20 | 8 | 91.40 (-0.02) | - | 67.58 (-0.01) | - |
| | 6 | 91.32 (-0.10) | - | 67.63 (+0.04) | - |
| | 5 | 91.36 (-0.06) | - | 67.46 (-0.13) | - |
| | 4 | 90.72 (-0.70) | 91.21 (-0.21) | 65.76 (-1.83) | 67.35 (-0.24) |
| ResNet-32 | 8 | 92.76 (+0.04) | - | 70.46 (+0.29) | - |
| | 6 | 92.62 (-0.10) | - | 70.08 (-0.09) | - |
| | 5 | 92.59 (-0.13) | - | 70.09 (-0.08) | - |
| | 4 | 91.73 (-0.99) | 92.35 (-0.37) | 68.24 (-1.93) | 69.38 (-0.79) |

The quantization results of AdderNets are reported in Table 1. For all networks, there is almost no accuracy loss in 8-bit quantization compared to the full-precision counterparts. For example, the accuracy of 8-bit quantized ResNet-32 is even 0.29% higher. For the 6-bit and 5-bit quantization, the accuracy loss exists, but is still within an acceptable range. For the more challenging 4-bit quantization, the accuracy loss of PTQ increases significantly. Therefore, QAT procedure is necessary in this case. After 50 epochs training, the accuracy loss was greatly reduced. For example, in the case of 4-bit ResNet-20 on CIFAR-100, the accuracy loss is reduced from 1.83% to 0.24%. If equipped with more refined QAT techniques, such as EWGS [22], KD [27] or simply a longer training epoch, the accuracy loss can be further reduced, but it is not the focus of this paper.

## 4.2 Experiments on ImageNet

**Re-train FP Networks.** We further conduct evaluation on ImageNet dataset [9], which contains over $1.2M$ training images and $50K$ validation images belonging to 1,000 classes. The pre-processing and data augmentation follow the same protocols as in [14]. We re-train two full-precision networks including ResNet-18 and ResNet-50 [14] according to the experimental settings in [4]. The entire training takes 150 epochs with a weight decay of $1 \times 10^{-4}$ and the other hyper-parameters is the same as that in CIFAR experiments. The final results are provided in the supplementary material and the baseline results of CNN and BNN are also cited from [4].

**Low-bit Quantization.** We adopt the same quantization settings discussed in CIFAR experiments except that the hyper-parameter $\alpha$ is set to $0.9992$. Besides, in the QAT procedure, we also keep the same settings as the full-training training, except that the learning rate is changed to $1 \times 10^{-3}$, and only 20 epochs are used for saving training time.

Table 2: Quantization results of AdderNets on ImageNet.

| Model | Bits | Top-1 Acc (%) | | Top-5 Acc (%) | |
|---|---|---|---|---|---|
| | | PTQ | QAT | PTQ | QAT |
| ResNet-18 | 8 | 67.7 (-0.2) | - | 87.7 (-0.1) | - |
| | 6 | 67.7 (-0.2) | - | 87.6 (-0.2) | - |
| | 5 | 67.4 (-0.5) | - | 87.3 (-0.5) | - |
| | 4 | 66.5 (-1.4) | 67.4 (-0.5) | 86.6 (-1.2) | 87.4 (-0.4) |
| ResNet-50 | 8 | 75.0 (-0.0) | - | 91.9 (-0.0) | - |
| | 6 | 75.0 (-0.0) | - | 91.8 (-0.1) | - |
| | 5 | 74.7 (-0.3) | - | 91.5 (-0.4) | - |
| | 4 | 73.2 (-1.8) | 73.7 (-1.3) | 90.7 (-1.2) | 90.9 (-1.0) |

The quantization results are reported in Table 2. For both networks, there is almost no loss in 8-bit and 6-bit quantization compared to the full-precision counterparts. For the 5-bit quantization, the accuracy loss is still within an acceptable range, *i.e.*, a maximum loss of 0.5%. Similar to CIFAR experiments, QAT is adopted in 4-bit quantization. After 20 epochs training, the accuracy loss was greatly reduced. For example, the accuracy loss of 4-bit ResNet-18 is reduced from 1.4% to 0.5%.

## 4.3 Comparisons with State-of-the-arts

**Image Classification.** We compare the proposed method with the existing AdderNet quantization technique QSSF [35] on image classification task. We first compare the PTQ results on ImageNet with it. The detailed Top-1 accuracy loss comparison with full-precision Adder ResNet-18 are reported in Figure 5 (a). QSSF [35] simply adopt only one shared scale to quantize both the weights and activations simultaneously, which ignores the properties of the distribution of the weights and activations of AdderNet, leading to the problems of "Over clamp" and "Bits waste", further leading to a poor accuracy. In contrast, we propose a quantization method consisting of three parts: clustering-based weights grouping, lossless range clamp for weights and outliers clamp for activations. The problems of "Over clamp" and "Bits waste" can be effectively resolved with our quantization method, further leading to a high quantized accuracy. As the number of bits decreases, the advantages of our method become more pronounced. For example, in the case of 4-bit, our method achieves 8.5% higher accuracy than QSSF [35].

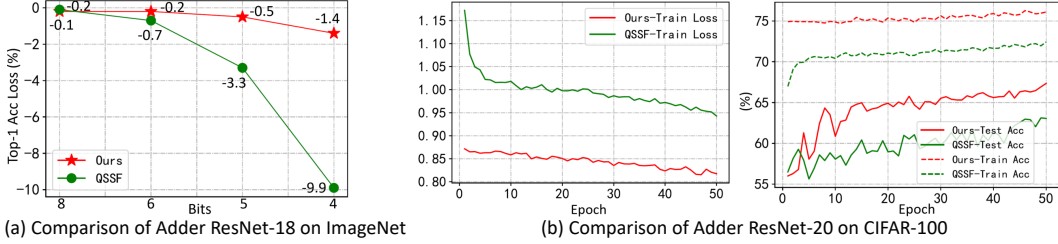

Figure 5: Comparisons with QSSF [35] on image classification task.

In addition, we also compare the QAT results on CIFAR-100 with it. In Figure 5 (b), we visualize the training loss, training accuracy and testing accuracy of the 4-bit Adder ResNet-20. For the accuracy curve, the solid lines and dash lines represent the testing accuracy and training accuracy, respectively. Our quantization method achieves a higher accuracy on both training and testing. Specifically, under the same epoch, our method achieves 67.35% test accuracy, which is higher than 64.61% achieved by QSSF [35].

**Object Detection.** AdderDet [5] presents an empirical study of AdderNet on object detection task. Besides, the PTQ mAP with one shared scale of 8-bit Adder FCOS [34] is also reported in this work, *i.e.*, 36.2 on COCO val2017 [26], which is 0.8 lower than the full-precision counterpart. To verify the generality of our quantization method on detection task, we conduct the PTQ experiment of the 8-bit Adder FCOS following the settings in AdderDet [5]. Specifically, the adder convolutions in the backbone and neck are quantized with 4-group shared scales, and the hyper-parameter $\alpha$ is set to 0.9992. The final 8-bit quantized mAP with our method is 36.5, which is 0.3 higher.

**Energy Cost.** In Figure 6, we compare the accuracy and energy cost of different network architectures after quantization, including AdderNet, CNN and ShiftAddNet [39]. The experiments are based on ResNet-20 network and CIFAR-100 dataset. We follow the related works [39, 35, 23] to measure the practical energy cost of ResNet-20 with various bits and basic operations on a FPGA platform. Besides, the quantized results of CNN is achieved by the BRECQ [24] PTQ method. The quantized accuracy of ShiftAddNet is cited from the original paper [39] and the energy of ShiftAddNet is calculated from the relative ratio of the energy of ShiftAddNet and AdderNet in the original paper [39], *i.e.*, -25.2%. From Figure 6, under similar energy cost, the quantized Adder ResNet-20 with our method achieves a higher accuracy.

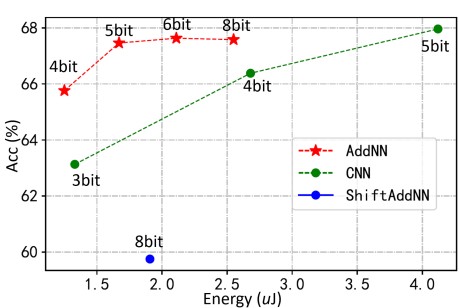

Figure 6: Comparisons of accuracy and energy cost of ResNet-20 on CIFAR-100.

## 4.4 Ablation Studies

In this subsection, we conduct thorough ablation studies based on the 4-bit quantized Adder ResNet-20 network on CIFAR-100 dataset.

**Analysis on Sub-Methods.** We first conduct analysis on three sub-methods as reported in Table 3, including group-shared scales, lossless range clamp for weights and outliers clamp for activations. The complete method achieves the highest accuracy of 67.35%, which is higher than the initial accuracy of 57.88%. Besides, the introduction of any sub-method contributes to the improvement of accuracy.

Table 3: Analysis on sub-methods.

| Method | | | |
|---|---|---|---|
| Group-Shared Scales | Weights Clamp | Activations Clamp | Acc (%) |
| | | | 57.88 |
| ✓ | | | 60.42 |
| | ✓ | | 62.30 |
| | | ✓ | 61.55 |
| ✓ | ✓ | | 65.17 |
| ✓ | ✓ | ✓ | 67.35 |

**Analysis on Group Number.** The results with various group number are reported in Table 4, and the relative FLOPs are calculated under the assumption that the number of channels of input and output is 32. From Table 4, when the group number is 4, the quantized model achieves the highest accuracy with a negligible increase in FLOPs. More group does not lead to a higher accuracy.

**Analysis on Group Methods.** We also conduct analysis on the group methods. The group number we used is 4, and the results are reported in Table 5. Uniform indicates that the weights are evenly grouped based on the output channel without clustering. Compared with other group methods, the clustering method based on $max(|W|)$ achieves the highest accuracy.

Table 4: Analysis on the group number.

| Group | Relative FLOPs | Acc (%) |
|---|---|---|
| 1 | 1 | 65.91 |
| 2 | 1.002 | 66.38 |
| 4 | 1.005 | 67.35 |
| 8 | 1.012 | 67.11 |

Table 5: Analysis on the group methods.

| Group Method | Acc (%) |
|---|---|
| Uniform | 65.72 |
| All | 66.61 |
| Mean | 66.86 |
| Max | 67.35 |

## 5 Conclusion

To further improve the efficiency of AdderNet, this paper studies the quantization algorithm for AdderNet. Considering that the weights of different channels may differ greatly, and the ranges of weights and activations are different, we propose a scheme of group-shared scales and a clamp strategy for weights and activations to improve the performance of quantized AdderNet. The effectiveness of the proposed quantization method for AdderNet is well-verified on several benchmarks. We achieve state-of-the-art results on various quantized network architectures and datasets. We believe that the advantages of low-bit quantized AdderNet can shed light into the design of low-power hardware for AI applications.

## Acknowledgement

We gratefully acknowledge the support of MindSpore [17], CANN(Compute Architecture for Neural Networks) and Ascend AI Processor used for this research. This research is supported by NSFC (62072449, 61972271); National Key R&D Program of China (2020YFC2004100); University of Macau Grant (MYRG2019-00006-FST).

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
