# Supplementary Material: Redistribution of Weights and Activations for AdderNet Quantization

**Ying Nie**[1]  **Kai Han**[1,2]  **Haikang Diao**[3]  **Chuanjian Liu**[1]  **Enhua Wu**[2,4]  **Yunhe Wang**[1]*
[1]Huawei Noah's Ark Lab
[2]State Key Lab of Computer Science, ISCAS & UCAS
[3]School of Integrated Circuits, Peking University [4]University of Macau
{ying.nie, kai.han, yunhe.wang}@huawei.com

## A  Appendix

### A.1  Proof of Proposition 2

*Proof.* For a weight $W \in \mathbb{R}^{d \times d \times c_{in} \times c_{out}}$ and an input activation $X \in \mathbb{R}^{h \times w \times c_{in}}$, the FLOPs of $\bar{X} \oplus \bar{W}$ is $2hw(c_{in}d^2 + 1)c_{out}$. In addition, the FLOPs required to calculate $\bar{w}$ and $\bar{x}$ are $c_{in}d^2c_{out}$ and $ghwc_{in}$, respectively. Besides, the size of the output activation can be calculated by:

$$h_{out} = \lfloor (h - d + 2 * padding)/stride + 1 \rfloor, w_{out} = \lfloor (w - d + 2 * padding)/stride + 1 \rfloor. \quad (1)$$

Without loss of generality, assume that the values of $padding$ and $stride$ are both 1, therefore, the FLOPs required to de-quantize the output activation is $h_{out}w_{out}c_{out} = (h - d + 3)(w - d + 3)c_{out}$. Finally, all FLOPs required for a layer quantization is:

$$FLOPs_{all} = 2hw(c_{in}d^2 + 1)c_{out} + c_{in}d^2c_{out} + ghwc_{in} + (h - d + 3)(w - d + 3)c_{out}. \quad (2)$$

Without loss of generality, assume that $d = 3$, $c_{in} = c_{out} = c$, and $h = w = k$, then

$$FLOPs_{all} = 18k^2c^2 + gk^2c + 9c^2 + 3k^2c. \quad (3)$$

Compared with only one scale ($g = 1$), FLOPs are increased by $r$ when adopting multiple scales ($g \geq 2$):

$$r = \frac{(g-1)k^2c}{18k^2c^2 + k^2c + 9c^2 + 3k^2c} = \frac{(g-1)k^2}{(18k^2 + 9)c + 4k^2} \approx \frac{g-1}{18c + 4}, \quad (4)$$

As we discussed in the section of experiments, the value of $g$ that we adopt is 4. In this case, Eq. 4 can be further simplified to

$$r \approx \frac{g-1}{18c+4} = \frac{3}{18c+4} \approx \frac{1}{6c+1}. \quad (5)$$

Considering that the magnitude of $c$ is generally in the tens or hundreds of common neural networks, thus the value of $r$ is very small. Therefore, the increase in FLOPs brought by the scheme of group-shared scales is negligible. □

### A.2  Full-precision Results

In the section of experiments, we re-trained multiple full-precision adder networks on various datasets. The full-precision results on CIFAR-10 and CIFAR-100 are reported in Table 1, and the full-precision results on ImageNet are reported in Table 2, both denoted by AddNN. The results of AddNN are basically consistent with the results in [1]. The baseline results of convolutional neural network (CNN) and binary neural network (BNN) are cited from [1].

---

*Corresponding author

36th Conference on Neural Information Processing Systems (NeurIPS 2022).

Table 1: Full-precision results on CIFAR-10 and CIFAR-100 datasets.

| Model | Method | # Mul. | # Add. | # XNOR. | CIFAR-10 (%) | CIFAR-100 (%) |
|---|---|---|---|---|---|---|
| VGG-Small | CNN | 0.65G | 0.65G | 0 | 93.80 | 72.73 |
| | BNN | 0.05G | 0.65G | 0.60G | 89.80 | 67.24 |
| | AddNN | 0.05G | 1.25G | 0 | 93.44 | 73.60 |
| ResNet-20 | CNN | 41.17M | 41.17M | 0 | 92.25 | 68.14 |
| | BNN | 0.45M | 41.17M | 40.72M | 84.87 | 54.14 |
| | AddNN | 0.45M | 81.89M | 0 | 91.42 | 67.59 |
| ResNet-32 | CNN | 69.12M | 69.12M | 0 | 93.29 | 69.74 |
| | BNN | 0.45M | 69.12M | 68.67M | 86.74 | 56.21 |
| | AddNN | 0.45M | 137.79M | 0 | 92.72 | 70.17 |

Table 2: Full-precision results on ImageNet.

| Model | Method | # Mul. | # Add. | # XNOR. | Top-1 Acc (%) | Top-5 Acc (%) |
|---|---|---|---|---|---|---|
| ResNet-18 | CNN | 1.8G | 1.8G | 0 | 69.8 | 89.1 |
| | BNN | 0.1G | 1.8G | 1.7G | 51.2 | 73.2 |
| | AddNN | 0.1G | 3.5G | 0 | 67.9 | 87.8 |
| ResNet-50 | CNN | 3.9G | 3.9G | 0 | 76.2 | 92.9 |
| | BNN | 0.1G | 3.9G | 3.8G | 55.8 | 78.4 |
| | AddNN | 0.1G | 7.6G | 0 | 75.0 | 91.9 |

## A.3 Analysis on the Ratio of Discarded Outliers

As we discussed in the subsection of outliers clamp for activations, the value $r_x = \widetilde{\mathbb{X}}[\lfloor \alpha * (n-1) \rfloor]$ is selected as the range of activations for the calculation of scale, where $\alpha \in (0, 1]$ is a hyper-parameter controlling the ratio of discarded outliers in activations. We supplement the ablation study of this ratio with 4-bit quantized adder ResNet-20 network on CIFAR-100 dataset.

Table 3: Analysis on the ratio of discarded outliers in activations.

| $\alpha$ | 0.9985 | 0.9990 | 0.9995 | 1.0 |
|---|---|---|---|---|
| Acc (%) | 67.29 | 67.35 | 67.11 | 65.17 |

As shown in Table 3, $\alpha = 1$ means that the scheme of outliers clamp for activations is not adopted, resulting in a significantly degraded quantized accuracy. The quantized accuracy can be improved with an appropriate $\alpha$.

## A.4 Quantization Results on Adder Vision Transformers

We also try the proposed quantization method on adder vision transformers [5]. We re-train the full-precision adder DeiT-T for 400 epochs from scratch on ImageNet dataset following [5], and the final top-1 accuracy of the full-precision adder DeiT-T is 68.3%. For the next quantization step, the number of groups we use is 4, the hyper-parameter $\alpha$ controlling the ratio of discarded outliers in activations is set to 0.9992. The accuracy drops after post-training quantization are reported in Table 4. The advantage of the proposed method over QSSFF [6] is significant. For example, at the case of W4A4, the accuracy drop of our method is 8.7%, which is much lower than the 16.3% of QSSF [6].

## A.5 Quantization Results on Lower-bit

We supplement the 3-bit PTQ quantization experiment of adder ResNet-20 on CIFAR-100 dataset. Besides, the comparisons with more CNN quantization methods are also supplemented. The detailed accuracy drops are reported in Table 5.

## A.6 Distribution of the Weights and Activations

In Figure 1, we visualize the histogram of the weights and activations in AdderNet. The input full-precision (FP) activations and weights in pre-trained AdderNet show a significant difference,

Table 4: Accuracy drops under various bits.

|  | W8A8(%) | W6A6 (%) | W4A4 (%) |
|---|---|---|---|
| Ours | -0.5 | -4.1 | -8.7 |
| QSSF [6] | -1.7 | -6.5 | -16.3 |

Table 5: Comparisons with more CNN quantization methods.

|  | W4A4 (%) | W3A3 (%) |
|---|---|---|
| AddNN | -1.83 | -6.02 |
| CNN AdaRound [4] | -2.01 | -6.77 |
| CNN BRECQ [3] | -1.74 | -5.95 |
| CNN QDROP [7] | -1.70 | -5.86 |

which pose a huge challenge for AdderNet quantization. Other AdderNet quantization methods [6, 2] fail to deal with this challenge, leading to the phenomenon of over clamp and bits waste, further resulting in a poor quantized accuracy. In contrast, our quantization method can effectively address this challenge by the redistribution of full-precision weights and activations, resulting in a good quantized accuracy. One-shared scale is adopted here for the simplification of visualization, and symmetric 4-bit quantization is taken as an example.

## A.7 Limitations and Societal Impacts

Our AdderNet quantization method has one major limitation: as the number of bits decreases, the accuracy loss of the quantization model will increase. Therefore, quantization-aware training is necessary for the low bits, which is time consuming and computationally consuming.

As for the societal impacts, the proposed quantization method can further reduce the energy consumption of AdderNet with a lower quantized accuracy loss. The low power devices equipped with quantized AdderNet can be deployed to surveillance scenario. If used improperly, there may be a risk of information leakage.

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

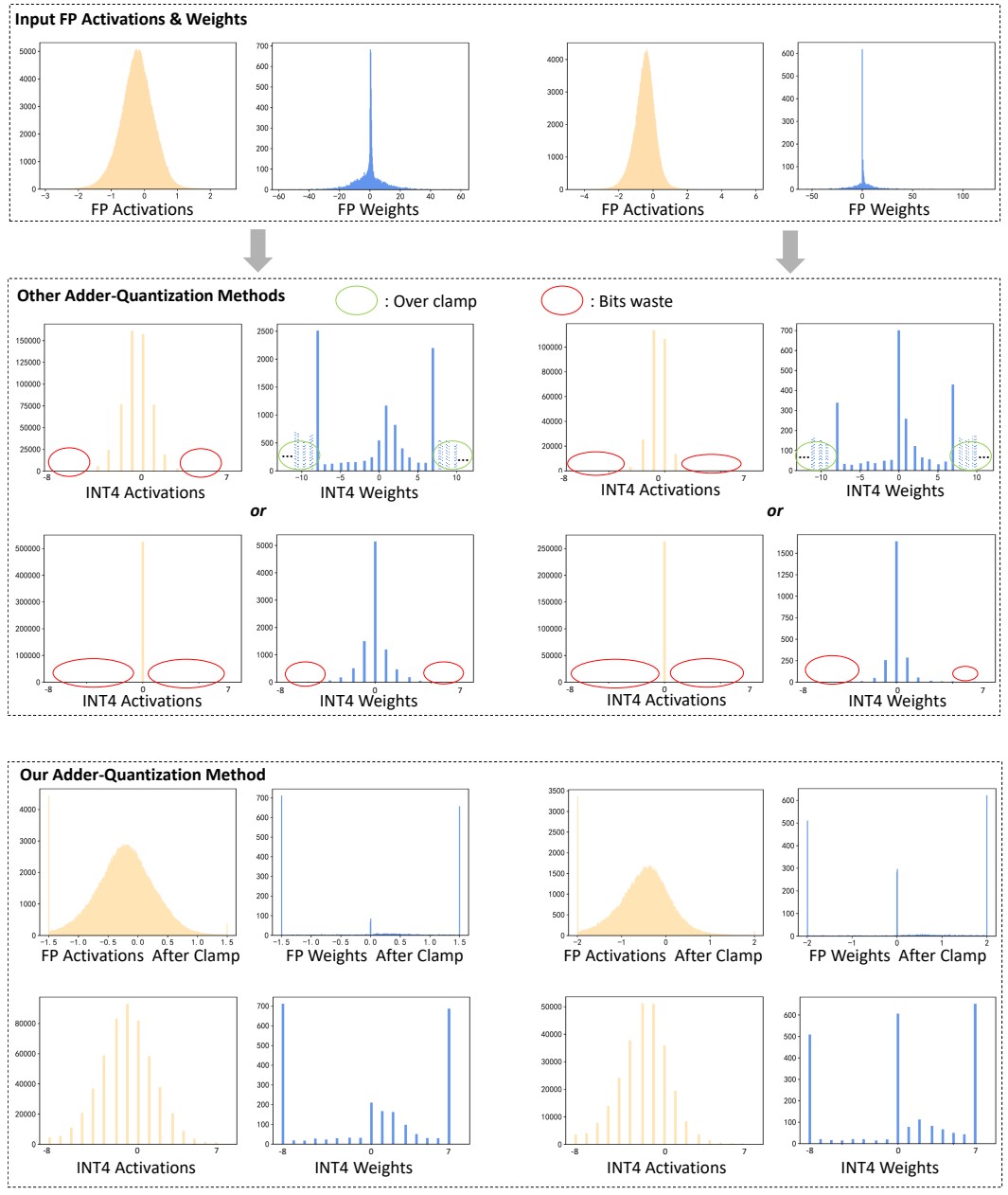

Figure 1: Distribution of the weights and activations in AdderNet.