# OpenReview forum: "Redistribution of Weights and Activations for AdderNet Quantization"
_NeurIPS.cc/2022/Conference — NeurIPS 2022 Accept_

### Official Review · Reviewer_QZcy · 2022-07-06

**Rating:** 8
**Confidence:** 5
**Soundness:** 3 good
**Presentation:** 3 good
**Contribution:** 3 good

**Summary:**

Quantization is an effective method to further reduce the energy consumption of AdderNets. However, previous AdderNets quantization methods cannot properly handle the challenge of large differences in weights and activations and often lead to a large accuracy degradation, especially in the case of low-bit (4-bit). This paper first reveals the key reasons for the poor accuracy of previous AdderNets quantization methods, namely “over clamp” and “bits waste”. Then a novel quantization method for AdderNets is proposed. Experiments on several datasets and models demonstrate the effectiveness of the proposed method.

**Questions:**

1. Is it a common phenomenon that the distribution of weights and activations in pre-trained adder neural networks differs greatly?
2. Can the authors further explain the meaning of the values ​​marked in red in Fig.3 (b) and Fig.3 (c)? How are these values ​​calculated?
3. The results in Fig.5 (a) are impressive, yet the authors are advised to make more discussions on the results to make the paper stronger.
4. Writing suggestion: Line 166: “both weights and features” should be “both weights and activations”, to be consistent with the full text.

**Limitations:**

The authors have discussed the limitations and potential negative societal impact of their work in Appendix.

**Strengths And Weaknesses:**

Strengths
1. The paper is extremely well structured and easy to follow, with motivation well-explained.
2. To my knowledge, this paper is by far the most comprehensive and systematic study of the quantization of AdderNets. Through thorough analysis, this paper concludes two main reasons for the poor accuracy of previous AdderNets quantization methods, namely “over clamp” and “bits waste”, which are insightful. The proposed scheme of the clustering-based weights grouping and the lossless range clamp for weights are interesting and novel.
3. Extensive experiments on different models and datasets. Superior performance compared to other AdderNets quantization methods. The thorough ablation studies verifies the effectiveness of each components. The distributions of weights and activations (Fig.1 in Appendix) demonstrate that the proposed method can effectively solve the problem of “over clamp” and “bits waste”, leading to a higher quantized performance.
Weaknesses
1. The values in Fig. 4 are too small to read. The authors are required to refine them.
2. The histogram for INT4 weights ​​adjacent to “over clamp” is significantly higher (Fig.1 in Appendix), however, this phenomenon is not expressed in the top of Fig.1 (c). The authors are advised to revise this detail for better presentation.

---

> ### Author Response · Authors · 2022-08-02
> **Response to Reviewer QZcy**
>
> Thanks for your strong support and detailed comments.
>
> **Q1:** The values in Fig. 4 are too small to read. The authors are required to refine them.
>
> **A1:** Thanks for the suggestion. This problem will be fixed in the updated manuscript.
>
> **Q2:** The histogram for INT4 weights adjacent to “over clamp” is significantly higher (Fig.1 in Appendix), however, this phenomenon is not expressed in the top of Fig.1 (c). The authors are advised to revise this detail for better presentation.
>
> **A2:** Thanks for pointing out this problem. The histograms closer to "over clamp" are indeed higher, because the out-of -range values are all truncated here. We will correct the Fig.1 (c) accordingly in the updated manuscript.
>
> **Q3:** Is it a common phenomenon that the distribution of weights and activations in pre-trained adder neural networks differs greatly?
>
> **A3:** Thanks for the good question. The full-precision AdderNet[1] on classification task first revealed this phenomena, that is, the absolute ranges of weights vary widely between output channels, and the absolute ranges of activations and weights also vary widely. Subsequent work[2] on detection task also revealed this phenomena. If some restrictions such as a higher L2 regularization or weight standardization[3] are added to weights druing training full-precision  AdderNet, the weight values can be effectively narrowed. However, it tends to result in lower accuracy of the full-precision AdderNet. We perform the experiments of full-precision adder ResNet-20 on CIFAR-100 dataset with various L2 regularization or weight standardization. Correspondingly, the mean and variance of the absolute values in the fully trained *layer1.1.conv2* are taken as an example to show the properties of weight values. The results are provided as follows:
>
> |             | 5e-4 (Our paper) | 1e-3  | 5e-3  |  WS   |
> | :-: | :--------------: | :---: | :---: | :---: |
> |   Acc (%)   |      67.59       | 55.61 | 41.23 | 61.52 |
> | Mean(\|W\|) |       6.05       | 2.13  | 0.26  | 0.65  |
> | Var(\|W\|)  |      90.33       | 38.27 | 17.70 | 0.58  |
>
> **Q4:** Can the authors further explain the meaning of the values marked in red in Fig.3 (b) and Fig.3 (c)? How are these values calculated?
>
> **A4:** Thanks for the nice concern. The value -1.6 marked in red in Fig.3 (b) denotes the value of activations after outliers clamping. Specifically, $\mathbb{X}= ${$1.4,1.2,1.1,4.5,0.0,0.6,0.4,0.8,1.6$}, then the sorted $\mathbb{\widetilde{X}}=${$0.0,0.4,0.6,0.8,1.1,1.2,1.4,1.6,4.5$}. We select the value$r_x=\mathbb{\widetilde{X}}[\lfloor0.9*8\rceil]=\mathbb{\widetilde{X}}[7]=1.6$ as the range of activations for the next calculation of scale. Correspondingly, any activations whose absolute value exceeds 1.6 will be clamped to 1.6 or -1.6, i.e., -4.5 clamped to -1.6.
>
> In Fig.3 (c), the values in $W_{3}$  are clamped to the range of activations after the step of outliers clamp for activations, i.e., 1.6. The bias term -126.2 marked in red is calculated according to Eq.16. That is, $-((16.8-1.6)+(12.3-1.6)+(15.4-1.6)+(11.3-1.6)+(21.0-1.6)+(19.2-1.6)+(10.0-1.6)+(18.4-1.6)+(16.2-1.6))=-126.2$
>
> **Q5:** The results in Fig.5 (a) are impressive, yet the authors are advised to make more discussions on the results to make the paper stronger.
>
> **A5:** Thanks for the nice suggestion. QSSF simply adopt only one shared scale to quantize both the weights and activations simultaneously, which ignores the properties of the distribution of the weights and activations of AdderNet, leading to the problems of "over clamp" and "bits waste", further leading to poor accuracy. In contrast, we propose a quantization method consisting of three parts: clustering-based weights grouping, lossless range clamp for weights and outliers clamp for activations. The problems of "over clamp" and "bits waste" can be effectively resolved with our quantization method, further leading to high quantized accuracy. Compared with QSSF, the advantage of our method is significant, especially at low bits. For example, in the case of 4-bit, our method achieves 8.5% higher accuracy than QSSF.
>
> **Q6:** Writing suggestion: Line 166: “both weights and features” should be “both weights and activations”, to be consistent with the full text.
>
> **A6:** Thanks for the suggestion. In the community of quantization, "activations" is indeed used more often than "features", we will correct to "both weights and activations" as the reviewer suggest.
>
> [1] Chen H, Wang Y, Xu C, et al. AdderNet: Do we really need multiplications in deep learning?[C]//Proceedings of the IEEE/CVF conference on computer vision and pattern recognition. 2020: 1468-1477.
>
> [2] Chen X, Xu C, Dong M, et al. An empirical study of adder neural networks for object detection[J]. Advances in Neural Information Processing Systems, 2021, 34: 6894-6905.
>
> [3] Qiao S, Wang H, Liu C, et al. Micro-batch training with batch-channel normalization and weight standardization[J]. arXiv preprint arXiv:1903.10520, 2019.

---

> > ### Comment · Reviewer_QZcy · 2022-08-09
> > **About the rebuttal**
> >
> > The authors' response has solved all of my concern, I will keep my rating about this work.

---

### Official Review · Reviewer_R6QH · 2022-07-11

**Rating:** 6
**Confidence:** 4
**Soundness:** 3 good
**Presentation:** 3 good
**Contribution:** 3 good

**Summary:**

This manuscript focuses on the problem of the quantization of AdderNet. The author has investigated the difference between AdderNet and traditional networks. Based on the differences, the dedicated quantization is achieved by redistributing the weights and activation of AdderNet. In the quantization method, three techniques are proposed to overcome the bit waste and over clamp problems, including clustering-based grouping quantization, range clamp of weights, and outlier clamp of activations. Experimental results show the effectiveness of the proposed method for quantizing AdderNet with different bit widths.

**Questions:**

Please see the weakness part.

**Limitations:**

Yes

**Strengths And Weaknesses:**

Pros:

- The manuscript is easy to follow. The analysis of the difference between conventional quantization methods for CNN and that for AdderNet is interesting. The statistics of the activations and weights of a pre-trained AdderNet are good.

- As a new kind of efficient neural network (AdderNet), how to effectively quantize it is a challenging problem. Quantization of such NN would put forward a faster and more energy-efficient model.

- The proposed method, includes clustering-based grouping quantization, range clamp of weights, and outlier clamp of activations, is promising for addressing the bit waste or over-clamp issues within AdderNet, which is also verified by the extension experiments.

Cons:

- Besides the FLOPs and energy, it would be great to report the inference time of the proposed method.

- It is highly recommended to add a more detailed recap about AdderNet, which will make the whole manuscript smoother, especially for those who do not familiar with it.

- In Fig. 3, how is the `-126.2` calculated? There is no detailed explanation about it.

- In line 45, “L1-norm quantization” is unclear. Does it mean an L1-norm-based quantization method or quantization for L1-norm operation?

Minor issues

-There is a strange “rec” symbol in Line 193

- There are several minor grammar issues. For example,  in line 21: “well-verified” should be “well verified”.

---

> ### Author Response · Authors · 2022-08-02
> **Response to Reviewer R6QH**
>
> Sincerely thanks for your constructive comments and support.
>
> **Q1:** Besides the FLOPs and energy, it would be great to report the inference time of the proposed method.
>
> **A1:** Thanks for the suggestion. For the modern CPU or GPU devices, a multiplication can be executed in a single cycle, which is roughly as fast as addition (https://stackoverflow.com/questions/21819682/is-integer-multiplication-really-done-at-the-same-speed-as-addition-on-a-modern).
>
> We measure the inference latency of full-precision AdderNets and CNNs on a single NVIDIA Tesla V100 GPU with 1x3x224x224 input. ResNet-18 architecture is adopted and the results are provided as follows. We can see that the latencies of CNN and AdderNet are very similar. To achieve higher speed on GPU,  more engineering works like professional CUDA implementation or dedicated hardware unit support, are required to handle the intensive adder operation.
>
> |   CNN (ms)    | AdderNet (ms) |
> | :-----------: | :-----------: |
> | $4.06\pm0.29$ | $4.18\pm0.35$ |
>
> **Q2:** It is highly recommended to add a more detailed recap about AdderNet, which will make the whole manuscript smoother, especially for those who do not familiar with it.
>
> **A2:** Thanks for the good suggestion. In L74-L86 in the paper, we provided a concise introduction and comparison of the computational paradigms of CNN and AdderNet. We will add more detailed recap on AdderNet and update the manuscript as the reviewer suggest.
>
> **Q3:** In Fig. 3, how is the -126.2 calculated? There is no detailed explanation about it.
>
> **A3:** Thanks for the question. The bias term -126.2 is calculated according to Eq.16, i.e., lossless range clamp for weights to the range of activations. Specifically, in Fig.3, $r_x=1.6$, then the bias term is equal to :$-((16.8-1.6)+(12.3-1.6)+(15.4-1.6)+(11.3-1.6)+(21.0-1.6)+(19.2-1.6)+(10.0-1.6)+(18.4-1.6)+(16.2-1.6))=-126.2$
>
> We will add detailed explanation where appropriate and update the manuscript accordingly.
>
> **Q4:** In line 45, “L1-norm quantization” is unclear. Does it mean an L1-norm-based quantization method or quantization for L1-norm operation?
>
> **A4:** Thanks for the nice concern. "L1-norm quantization" means the quantization for L1-norm operation (Eq.2). We will fix this ambiguity in the updated manuscript.
>
> **Q5:** The strange “rec” symbol in Line 193.
>
> **A5:** Thanks for the nice concern. The "rec" symbol in L193 indicates the end of the proof of Theorem 1, which is in line with mathematical norms.
>
> **Q6:** There are several minor grammar issues.
>
> **A6:** Thanks for pointing out the minor grammar issues in the paper. These issues will be corrected as the reviewer suggest in the updated manuscript.

---

### Official Review · Reviewer_zyyr · 2022-07-12

**Rating:** 4
**Confidence:** 4
**Soundness:** 2 fair
**Presentation:** 3 good
**Contribution:** 2 fair

**Summary:**

The paper proposes a new quantization scheme for Addernets. Specifically, the authors propose to cluster model weights in the channel dimension, where each cluster is assigned its own scale value for quantization. This ensures that the scales can better represent the range of values, which may be different for different weight channels. The authors further propose to absorb the error caused by clamping the weights inside the layer bias, which helps restore accuracy. Finally, the proposed method removes outliers when quantizing the activations, therefore tailoring the scale to better represent the valid range of data.

**Questions:**

Questions:
- It appears that the models are not trained with a weight regularization term which leads to a wide range of weight values in Figure 2. A standing question is whether adding strict regularization to training can change the effectiveness of the proposed method? In that scenario, the weights are forced to be within certain bounds, which may reduce the benefits of clustering as the same scale may likely work for all weights.

Limitations:
- The I term in equation 13 of the paper merely depends on the weight values and not the activations. Therefore, while the derived scale is perhaps optimized for the weights, there is no guarantee that it will work for the range of activations. The accuracy gains of the proposed method may therefore be due to the more fine-grained weight scales, rather than the fact that the scales are actually tailored to both the activation and the weights, which is what is assumed after reading the introduction.

**Limitations:**

The authors have not discussed the limitations or potential negative social impact of their work.

**Strengths And Weaknesses:**

Strengths:
- The paper is well-written and the ideas are clearly explained.
- The proposed method significantly improves the accuracy after quantization, compared to prior methods.

Weaknesses:
- Some of the claims are not backed up by the method. Specifically, the authors mention that a shortcoming of prior work is using the same scale for weights and activations which is decided based on either of the two and therefore may not best fit the other. The proposed method also adopts the same scheme, where the scales are still determined by either the weights or the activations with the only difference being the increased granularity of the scale choices due to the channel clustering. Please find more details on this in the next section.
- Some questions remain regarding applying the method to new models, e.g., how to determine the number of clusters for new benchmarks.

---

> ### Author Response · Authors · 2022-08-02
> **Response to Reviewer zyyr (Part 2/2)**
>
> **Q5:** The accuracy gains of the proposed method may be due to the more fine-grained weight scales, rather than the fact that the scales are actually tailored to both the activation and the weights.
>
> **A5:** Thanks for the nice concern. In the above question, we explained that the scales are actually derived by taking both weights and activations into account. Clustering-based weights grouping, i.e., more fine-grained weight scales, is only a part of our method. Our technical contributions include 1) distribution analysis of AdderNet, 2) group-shared quantization scales, 3) clustering-based weights grouping, 4) lossless range clamp for weights, and 5) outliers clamp for activations. The ablation study on sub-methods is shown in Tab.3 in the paper. Here we supplement the ablation study on ImageNet dataset with adder ResNet-18 architecture to better clarify the effectiveness of each part of our method. The 4-bit post-training quantized top-1 accuracy is reported below:
>
> | Group-shared scales | Weights clamp |  Acts clamp  |   Acc (%)   |
> | :-----------------: | :-----------: | :----------: | :---------: |
> |                     |               |              |    58.0     |
> |    $\checkmark$     |               |              | 61.7 (+3.7) |
> |                     | $\checkmark$  |              | 63.2 (+5.2) |
> |                     |               | $\checkmark$ | 61.0 (+3.0) |
> |                     | $\checkmark$  | $\checkmark$ | 64.9 (+6.9) |
> |    $\checkmark$     | $\checkmark$  |              | 65.3 (+7.3) |
> |    $\checkmark$     | $\checkmark$  | $\checkmark$ | 66.5 (+8.5) |
>
> **Q6:** The authors have not discussed the limitations or potential negative social impact of their work.
>
> **A6:** We have discussed the limitations and potential negative social impact of our work in A.5 in the supplementary material. For more details, please refer to this section.
>
> [1] Qiao S, Wang H, Liu C, et al. Micro-batch training with batch-channel normalization and weight standardization[J]. arXiv preprint arXiv:1903.10520, 2019.

---

> ### Author Response · Authors · 2022-08-02
> **Response to Reviewer zyyr (Part 1/2)**
>
> We would like to sincerely thank the reviewer for providing a constructive review and detailed comments.
>
> **Q1:** Some of the claims are not backed up by the method. Specifically, the authors mention that a shortcoming of prior work is using the same scale for weights and activations which is decided based on either of the two and therefore may not best fit the other. The proposed method also adopts the same scheme, where the scales are still determined by either the weights or the activations with the only difference being the increased granularity of the scale choices due to the channel clustering.
>
> **A1:** Thanks. Firstly, please allow us to correct one point: what our paper claims is that the prior works simply adopt only one shared scale to quantize both the weights and activations simultaneously (L8, L46-L47), which ignores the properties of the distribution of the weights and activations of AdderNet, leading to the problems of "over clamp" and "bits waste", further leading to poor quantized accuracy. Secondly, as we emphasized in paper, due to the limitation that the commutative law in multiplication does not hold in adder operation, non-shared scales cannot be adopted when quantizing the weights and activations in adder convolution (Eq.7). Therefore, the target of our work is how to solve the problems of "over clamp" and "bits waste" caused by adopting only one shared scale in prior works under the restriction that only the same scales between weights and activations can be adopted, and finally improve the accuracy of quantized AdderNet. Our method consists of three parts: clustering-based weights grouping, lossless range clamp for weights and outliers clamp for activations. Clustering-based weights grouping is only one part of our method. From Tab.3 in paper, using weights grouping alone can improve the performance, and combining all the proposed components obtains a much higher accuracy.
>
> **Q2:** Some questions remain regarding applying the method to new models, e.g., how to determine the number of clusters for new benchmarks.
>
> **A2:** The number of clusters is set empirically. Generally, when the number of clusters is equal to 4, the quantized AdderNets tend to achieve higher accuracy. From the Tab.4 in paper, more groups do not necessarily result in higher accuracy.
>
> **Q3:** It appears that the models are not trained with a weight regularization term which leads to a wide range of weight values in Figure 2. A standing question is whether adding strict regularization to training can change the effectiveness of the proposed method? In that scenario, the weights are forced to be within certain bounds, which may reduce the benefits of clustering as the same scale may likely work for all weights.
>
> **A3:** It's an interesting question. Adding some tricks like a higher L2 regularization or weight standardization[1] can effectively narrow the range of weight values. However, it tends to result in lower accuracy of the full-precision AdderNet, let alone the quantized accuracy. We perform the experiments of full-precision adder ResNet-20 on CIFAR-100 dataset with various L2 regularization or weight standardization. Correspondingly, the mean and variance of the absolute values in the fully trained *layer1.1.conv2* are taken as an example to show the properties of weight values. The results are provided as follows:
>
> |             | 5e-4 (Our paper) | 1e-3  | 5e-3  |  WS   |
> | :---------: | :--------------: | :---: | :---: | :---: |
> |   Acc (%)   |      67.59       | 55.61 | 41.23 | 61.52 |
> | Mean(\|W\|) |       6.05       | 2.13  | 0.26  | 0.65  |
> | Var(\|W\|)  |      90.33       | 38.27 | 17.70 | 0.58  |
>
> **Q4:** The $\mathbb{I}$ term in equation 13 of the paper merely depends on the weight values and not the activations. Therefore, while the derived scale is perhaps optimized for the weights, there is no guarantee that it will work for the range of activations.
>
> **A4:** Thanks for the nice concern. As the reviewer said, the $\mathbb{I}$ term in Eq.13 is calculated merely depends on the weights and not the activations. However, this does not mean that the derived scale is only optimized for weights without taking activations into account. As we stated in L161-L167 of the paper, for the majority groups where the range of weights exceeds the range of activations, the scheme of lossless range clamp for weights is adopted (Fig.3 (c)). That is, the weights is clamped to the range of activations and the error caused by clamping weights is absorbed inside the layer bias, then the scale is derived depends on the range of activations. For the minority groups where the range of weights is within the range of activations, the difference between the range of weights and activations is usually small, so the scale derived based on the range of weights can be adopted for quantizing both the weights and activations. In summary, the derived scales take both weights and activations into account.

---

### Official Review · Reviewer_2x7U · 2022-07-14

**Rating:** 6
**Confidence:** 4
**Soundness:** 3 good
**Presentation:** 3 good
**Contribution:** 3 good

**Summary:**

This paper first thoroughly analyzes the difference in distributions of weights and activations in AdderNet and then proposes a new quantization algorithm by redistributing the weights and the activations.

**Questions:**

1. The authors show the quantization results from 8bit to 4bit, and I am curious about the accuracy on lower-bit quantization, e.g., 3bit?
Fig. 6 shows energy and accuracy comparisons with CNN quantized via BRECQ PTQ method. Could you show comparisons with more CNN quantization methods?

2. Is the quantization bit of output (before adding the bias) consistent with that of input and weight? Do you also quantize gradients and errors during training?

3. The authors cluster weights in AdderNet into several groups, and then leverage different inter-group scale factors to quantize both weights and input. By doing this, I am concerned about whether it will consume more energy to transfer the aforementioned different quantized inputs when deployed onto mobile devices.

4. Do you have any plan to open source the kernel implementation of AdderNet and corresponding quantization schemes?

---
After reading the author's post, most of my questions are resolved and I am tending to accept it. The new experiments or tables can be added to the revision.

**Limitations:**

The comparison with CNN quantization method seems to be not very adequate.

**Strengths And Weaknesses:**

Strengths:
1. This paper conducts a thorough study of the dilemma in AdderNet quantization, and proposes an effective method to solve this problem.
2. The paper is clearly presented.
3. I am glad to see that the accuracy drop is within 2% for ImageNet even at 4 bits.

Weaknesses (suggestions):

1. The accuracy and energy comparisons with quantized CNN seem to be not very adequate. Better to compare the accuracy drops in quantized CNNs as well as the currently presented accuracy drop in AdderNet. So the reader can get the full information whether AdderNet is more advanced as compared to CNNs in terms of quantization.

2. AdderNet is a specific neural network, it is not clear whether the proposed methods can be generalized to other neural networks will similar distribution properties.

3. Only classification results are shown, how about other downstrem tasks, e.g., detection, segemetation. Or other network architectures, e.g., ViTs with adder kernels.

4. Does the AdderNet compatiable with SSL pretraining? e.g., MAE pretraining, and how the quantization scheme different for pretraining stage and fine-tuning or normal training stages?

5. How is the latency or throughput in real devices? I am curious of this since there is no available efficient CUDA implementation of AdderNet open-sourced as of now. Also, quantization is also not efficient for general CPU/GPU devices. Have you deployed the trained models to real devices yet?

I am open to further boost the score or champion this paper if the rebuttal is also sound and can somewhat solve my quesions/concerns.

---

> ### Author Response · Authors · 2022-08-02
> **Response to Reviewer 2x7U (Part 2/2)**
>
> **Q5:** How is the latency or throughput in real devices?  Have you deployed the trained models to real devices yet?
>
> **A5:** For the modern CPU or GPU devices, a multiplication can be executed in a single cycle, which is roughly as fast as addition (https://stackoverflow.com/questions/21819682/is-integer-multiplication-really-done-at-the-same-speed-as-addition-on-a-modern).
>
> We measure the inference latency of full-precision AdderNets and CNNs on a single NVIDIA Tesla V100 GPU with 1x3x224x224 input. ResNet-18 architecture is adopted and the results are provided as follows. We can see that the latencies of CNN and AdderNet are very similar. To achieve higher speed on GPU,  more engineering works like professional CUDA implementation or dedicated hardware unit support, are required to handle the intensive adder operation.
>
> |   CNN (ms)    | AdderNet (ms) |
> | :-----------: | :-----------: |
> | $4.06\pm0.29$ | $4.18\pm0.35$ |
>
> The most important benefits of AdderNet are reducing energy and circuit resources. These advantages of AdderNet will be maximally utilized by specific hardware implementation such as FPGA and ASIC. For example, QSSF[3] has deployed the quantized AdderNet on FPGA platform. Compared to 8-bit CNN, the logic resource utilization and energy consumption of 8-bit AdderNet can be reduced by 61.9% and 56.6%, respectively.
>
> **Q6:** How about the accuracy on lower-bit quantization, e.g., 3-bit? Fig. 6 shows ... Could you show comparisons with more CNN quantization methods?
>
> **A6:** We supplement the 3-bit PTQ quantization experiment of adder ResNet-20 on CIFAR-100 dataset. Besides, the comparisons with more CNN quantization methods are also supplemented. The detailed accuracy drops are reported below:
>
> |                 | W4A4 (%) | W3A3 (%) |
> | :-------------: | :------: | :------: |
> |      AddNN      |  -1.83   |  -6.02   |
> | CNN AdaRound[4] |  -2.01   |  -6.77   |
> |  CNN BRECQ[5]   |  -1.74   |  -5.95   |
> |  CNN QDROP[6]   |  -1.70   |  -5.86   |
>
> QDROP[6] establishes a SOTA result for PTQ of CNNs, as far as we know. The difference in the quantization method of CNN basically does not affect its energy consumption, which has been reported in the first question above. The accuracy drop of the the quantized AdderNet is slightly higher when compared with CNN QDROP and CNN BRECQ, but the energy advantage of the quantized AdderNet is more pronounced.
>
> **Q7:** Is the quantization bit of output (before adding the bias) consistent with that of input and weight? Do you also quantize gradients and errors during training?
>
> **A7:** The output bits are usually higher than input  and weight to prevent overflow of intermediate results, which is also a common practice for quantized CNNs. Our work focuses on the quantizing of weights and activations for inference. Quantizing gradients and errors is another topic of training acceleration, which is not covered in this paper.
>
> **Q8:** The authors cluster weights in AdderNet into several groups, ... I am concerned about whether it will consume more energy to transfer the aforementioned different quantized inputs when deployed onto mobile devices.
>
> **A8:** 1) The main computational cost of neural network relies on the matrix multiplications. The increased FLOPs introduced by the multiple inter-group scale factors are negligible as presented in L172-L173 of our paper, the detailed proof is presented in A.1 in the supplementary material. 2) In addition, we only need to quantize the activations during inference since the weights can be quantized offline and saved. The proposed activation quantization is an elementwise operation, which does not affect the most energy-intensive convolution operation. Therefore, multiple inter-group scale factors have little effect on the overall energy consumption.
>
> **Q9.** Do you have any plan to open source the kernel implementation of AdderNet and corresponding quantization schemes?
>
> **A9:** Yes, we are glad to open-source the kernel implementation of AdderNet and the corresponding quantization methods, and we hope that our work on quantization of AdderNet can bring some contributions to the multiplication-less neural network and the quantization community.
>
> [1] Chen X, et al. An empirical study of adder neural networks for object detection[J]. NeurIPS, 2021.
>
> [2] Shu H, et al. Adder attention for vision transformer[J]. NeurIPS, 2021.
>
> [3] Wang Y, et al. AdderNet and its minimalist hardware design for energy-efficient artificial intelligence[J]. arXiv, 2021.
>
> [4] Nagel M, et al. Up or down? adaptive rounding for post-training quantization[C]. ICML, 2020.
>
> [5] Li Y, et al. Brecq: Pushing the limit of post-training quantization by block reconstruction[J]. arXiv, 2021.
>
> [6] Wei X, et al. QDrop: Randomly Dropping Quantization for Extremely Low-bit Post-Training Quantization[J]. arXiv, 2022.

---

> ### Author Response · Authors · 2022-08-02
> **Response to Reviewer 2x7U (Part 1/2)**
>
> We would like to sincerely thank the reviewer for providing a constructive review and detailed comments.
>
> **Q1:** Better to compare the accuracy drops in quantized CNNs as well as the currently presented accuracy drop in AdderNet.
>
> **A1:** Thanks for the nice suggestion. We adopt the BRECQ PTQ method and further conduct the PTQ experiments on CIFAR-100 dataset with CNN ResNet-20 architecture.  The comparison of accuracy drops are reported in the following table:
>
> |       | W8A8 (%) | W6A6 (%) | W5A5 (%) | W4A4 (%) |
> | :-: | :-: | :-: | :-: | :-: |
> |  CNN  |  +0.02   |  +0.02   |  -0.11   |  -1.74   |
> | AddNN |  -0.01   |  +0.04   |  -0.13   |  -1.83   |
>
> The corresponding energy consumptions are also calculated under various bits, and the results are reported below:
>
> |       | W8A8 ($\mu$J) | W6A6 ($\mu$J) | W5A5 ($\mu$J) | W4A4 ($\mu$J) |
> | :-: | :-: | :-: | :-: | :-: |
> |  CNN  |     9.47      |     7.11      |     4.13      |     2.75      |
> | AddNN |     2.55      |     2.02      |     1.65      |     1.33      |
>
> From the above comparisons of accuracy drop and energy consumption, AdderNets achieve a comparable accuracy drop with CNN, but with noticeable lower energy consumption.
>
> **Q2:** It is not clear whether the proposed methods can be generalized to other neural networks with similar distribution properties.
>
> **A2:** ShiftAddNet is another impressive multiplication-less neural network that involves shift and add operations.  The weights and activations of ShiftAddNet exhibit similar distributions to those of AdderNet. We apply the proposed method to quantize the add operation in ShiftAddNet ResNet-20 on CIFAR-100 dataset.  Specifically, the number of groups we use is 4, the hyper-parameter $\alpha$ controlling the ratio of discarded outliers in activations is set to 0.999. In addition, only PTQ is adopted within limited time. The accuracy drop under various bits is reported below:
>
> | W8A8 (%) | W6A6 (%) | W5A5 (%) | W4A4 (%) |
> | :-: | :-: | :-: | :-: |
> |  +0.02   |  -0.03   |  -0.41   |  -1.25   |
>
> **Q3:** How about other downstrem tasks, e.g., detection, segmentation. Or other network architectures, e.g., ViTs with adder kernels.
>
> **A3:** Thanks for the suggestions.  We have performed the experiment on detection task in L266-L272 in the paper.  Specifically, the mAP of 8-bit quantized Adder FCOS on COCO val2017 dataset with our quantization method is 36.5, which is 0.3 higher than the quantization method in AdderDet[1].
>
> As for the segmentation task, as far as we know, there is no work on applying full-precision AdderNet to segmentation task so far. Since our work is focused on the quantization of AdderNet, it is difficult for us to first train a full-precision AdderNet on segmentation task with limited time and no successful precedent.  However, we think this is a good topic that can be explored in the future.
>
> As for other network architectures,  there is indeed a very interesting work, i.e., AdderViT[2]. We reproduce the full-precision adder DeiT-T on ImageNet dataset following AdderViT[2]. Specifically, to save training time, we only train 400 epochs from scratch instead of the 600 epochs in AdderViT. The top-1 accuracy of our full-precision adder DeiT-T on ImageNet dataset is 68.3%. The weights and activations of adder operation in AdderViT exhibit similar distributions to those of CNN AdderNet. Therefore, the proposed quantization method can be adopted to quantize the weights and activations involved in the adder linear transformation and adder multi-head self-attention in AdderViT.  Specifically, the number of groups we use is 4, the hyper-parameter $\alpha$ controlling the ratio of discarded outliers in activations is set to 0.9992, only PTQ is adopted  here due to limited time. Note that the PTQ accuracy can be further improved with our QAT method. The top-1 accuracy drop under various bits is reported below:
>
> |         | W8A8 (%) | W6A6 (%) | W4A4 (%) |
> | :-: | :-: | :-: | :-: |
> |  Ours   |   -0.5   |   -4.1   |   -8.7   |
> | QSSF[3] |   -1.7   |   -6.5   |  -16.3   |
>
> The advantage of our method over QSSF[3] is significant. For example, at W4A4, the accuracy drop achieved by our method is 8.7%, which is much lower than the 16.3% of QSSF.
>
> **Q4:** Does the AdderNet compatiable with SSL pretraining? e.g., MAE pretraining, and how the quantization scheme different for different stages?
>
> **A4:** Thanks for the insightful questions. The full-precision AdderNet has made impressive progress across different networks including CNNs and Transformers. Therefore, we believe that AdderNet is also likely to achieve some interesting results for MAE pretraining. However, this is far beyond the scope of our paper, and we cannot give detailed experimental results in a limited time. Overall, AdderNet has plenty of room for exploration, and we hope that our work on quantization of AdderNet can shed some light on the multiplication-less neural network and the quantization community.

---

### Meta-Review · Area_Chair_ZMQW · 2022-08-21

**Recommendation:** Accept
**Confidence:** Certain

**Metareview:**

The reviewers were mostly positive about this paper [8,6,6,4], while the negative reviewer did not update the review or respond after the author's response. I do not see any major issues remaining. The suggested method seems interesting, novel, and achieves good empirical results.

**Award:**

No

---

### Decision · Program_Chairs · 2022-09-14

Accept